# Phenotypes and Serum Biomarkers in Sarcoidosis

**DOI:** 10.3390/diagnostics14070709

**Published:** 2024-03-27

**Authors:** Matteo Della Zoppa, Francesco Rocco Bertuccio, Ilaria Campo, Fady Tousa, Mariachiara Crescenzi, Sara Lettieri, Francesca Mariani, Angelo Guido Corsico, Davide Piloni, Giulia Maria Stella

**Affiliations:** 1Pneumology Unit, IRCCS Policlinico San Matteo Foundation, Viale Golgi 19, 27100 Pavia, Italy; matteo.dellazoppa@gmail.com (M.D.Z.); francesco.bertuccio01@gmail.com (F.R.B.); fady.tousa01@universitadipavia.it (F.T.); mariachiara.crescenzi01@universitadipavia.it (M.C.); sara.lettieri01@universitadipavia.it (S.L.); fr.mariani@smatteo.pv.it (F.M.); a.corsico@smatteo.pv.it (A.G.C.); d.piloni@smatteo.pv.it (D.P.); g.stella@smatteo.pv.it (G.M.S.); 2Department of Internal Medicine and Medical Therapeutics, University of Pavia Medical School, 27100 Pavia, Italy

**Keywords:** sarcoidosis, phenotyping, interstitial lung disease, lung fibrosis, inflammatory granulomatous diseases

## Abstract

Sarcoidosis is a multisystem disease, which is diagnosed on a compatible clinical presentation, non-necrotizing granulomatous inflammation in one or more tissue samples, and exclusion of alternative causes of granulomatous disease. Considering its heterogeneity, numerous aspects of the disease remain to be elucidated. In this context, the identification and integration of biomarkers may hold significance in clinical practice, aiding in appropriate selection of patients for targeted clinical trials. This work aims to discuss and analyze how validated biomarkers are currently integrated in disease category definitions. Future studies are mandatory to unravel the diverse contributions of genetics, socioeconomic status, environmental exposures, and other sociodemographic variables to disease severity and phenotypic presentation. Furthermore, the implementation of transcriptomics, multidisciplinary approaches, and consideration of patients’ perspectives, reporting innovative insights, could be pivotal for a better understanding of disease pathogenesis and the optimization of clinical assistance.

## 1. Introduction

Sarcoidosis is an heterogenic disease characterized by the multisystemic involvement of non-caseating epithelioid cell granulomas and a clinically coherent presentation. Diagnosis confirmation may necessitate the exclusion of alternative causes of other granulomatous diseases [1,2]. More than half of the populations affected by sarcoidosis have entered the sixth decade of life, and there is a female/male ratio of 1.2 to 1.5:1. The prevalence of the disease is estimated to be between 1 and 40 cases per 100,000 individuals [3,4]. The organs most commonly involved are the lungs, lymph nodes, skin, and eyes [5,6]. Despite the inherently unpredictable nature of sarcoidosis, a substantial proportion of patients, up to two-thirds, may undergo spontaneous remission. Conversely, the remaining one-third of patients with disease remaining often develop chronic, progressive disease, occasionally presenting with life-threatening features [7]. The wide variability in clinical presentations and their evolution makes it necessary to identify clusters of patients with similar clinical characteristics and a similar course of disease, which are defined as phenotypes of sarcoidosis [8]. According to the heterogeneity in clinical presentation, the prognosis is very variable: the rate of spontaneous remission varies significantly based on the disease stage, reaching up to 80% in patients at stage I, but markedly diminishing to nearly 0% in stage IV. Patients require systemic treatment in varying proportions, from 10 to 73%, in different cohorts [9,10,11]. Because of this, the clinical manifestations, involved organs, severity of disease, and underlying conditions necessitate an individualized therapeutical strategy. The development of respiratory symptoms is the most common indication for treatment, followed by extra-pulmonary involvement of the cardiovascular, cutaneous, and nervous systems.

Currently, the treatment for sarcoidosis includes first-line use of corticosteroids, followed by cytotoxic drugs as second-line treatment, and anti-tumor necrosis factor (TNF) biologics as third-line agents. Rituximab and repository corticotropin injections have been recently examined, with encouraging results [1,2]. In 2021, a multidisciplinary task force established by the European Respiratory Society (ERS) formulated eight clinical questions using the PICO (Patients, Intervention, Comparison, Outcomes) framework, concentrating on the management of sarcoidosis. Recommendations were developed in accordance with the GRADE (Grading of Recommendations Assessment, Development and Evaluation) methodology. The committee conducted an extensive review of therapeutic approaches for pulmonary, cutaneous, cardiac, and neurologic manifestations, besides considerations for fatigue and small fiber neuropathy. However, specific conclusions regarding dosing, monitoring, and treatment duration for any diagnostic category were not reached [2,12]. In this review, we aim at summarizing and discussing the known diagnostic and prognostic biomarkers with the purpose of identifying phenotypes and treatable traits.

## 2. Phenotyping the Disease

Phenotypic classifications aim to predict individual patient outcomes, guide specific therapeutic interventions and assist clinicians in the management of patients, or refer them to experienced centers [13,14,15]. In 1960, Karl Wurm first proposed a phenotypical staging of sarcoidosis [16], and it was modified by Guy Scadding in 1961 [17]. This classification that was based entirely on chest X-ray findings showing involvement of the lungs and hilar lymph nodes. However, Scadding observed that certain extrathoracic features were not associated with radiographic patterns. Furthermore, subsequent studies revealed a weak correlation with disease severity, pulmonary function tests, or the necessity for treatment. [17,18,19,20,21]. Conversely, from the literature it is evident that there is a positive correlation with factors such as gender, ethnicity, respiratory symptoms, pulmonary function tests, high-resolution CT (HRCT) scores, bronchial granuloma density, prognosis, and diagnostic delay [22,23,24,25,26,27].

### 2.1. Systemic Disease

In 2019, Perez-Alvarez et al. conducted an analysis involving 1230 patients to investigate the association between systemic sarcoidosis phenotype and radiological stages. According to the authors, epidemiologic profile, extrathoracic involvement, and initial treatment were mainly associated with the presence or absence of pulmonary involvement rather than Scadding’s classification. They also found that pulmonary involvement represented by Scadding’s radiologic stages was associated with a different systemic phenotype at diagnosis, with a lower frequency of systemic disease but a higher risk of concomitant abdominal involvement (liver, spleen). In opposition, the absence of pulmonary involvement is more frequently associated with systemic disease, in particular with an increased risk of Lofgren’s features, such as cutaneous/musculoskeletal disease and cephalic involvement (ENT, ocular, neurological) [28].

In 2018, a multicentric study published in a European respiratory journal proposed a classification of systemic sarcoidosis, identifying five clusters of patients based on organ involvement: (I) abdominal, (II) ocular–cardiac–cutaneous–CNS, (III) musculoskeletal–cutaneous (IV) pulmonary/lymph nodal (V) extrapulmonary sarcoidosis. According to the authors, those classification should provide a better definition of the subcohort of patients with sarcoidosis [29].

Phenotyping is most effective when thoracic organs are analyzed individually, rather than grouped according to Scadding stages. This distinction is critical because Scadding stage classification involves organs with opposite prognoses, leading to alternative therapeutic management, and their involvement is distributed across stages. In this cohort, 8% of patients have sarcoidosis limited to extrathoracic organs at the time of diagnosis (Scadding stage 0), and nearly 50% of these patients have isolated involvement of only one extrathoracic organ. This cohort shows a specific phenotype characterized by a predominance of women diagnosed at on older age and a higher frequency of all extrathoracic organ-specific involvement (except muscle and kidney); this is not a surprising finding since diagnosis in these patients is based on extrathoracic disease [28].

### 2.2. Pulmonary Disease

Lung involvement in sarcoidosis is the predominant presentation and chest X-ray abnormality should be observed in 80% of cases [9]. As previously reported, pulmonary involvement should exhibit a markedly different presentation and clinical course. Some patients may experience an asymptomatic disorder, occasionally discovered incidentally, while others may develop severe, progressive lung failure requiring lung transplantation. According to a recent review, the mortality rate in pulmonary sarcoidosis is estimated to be approximately 7% of patients with a 5-year follow-up [9]. Several studies have identified factors that may be used as predictors for a patient’s prognosis in pulmonary sarcoidosis, including reduction in forced vital capacity (FVC), pulmonary arterial hypertension, and radiographic presence of pulmonary fibrosis. Despite these efforts, attempts to formulate a reliable prognostic algorithm for pulmonary involvement have proven largely unsuccessful. This challenge has, in turn, hindered the establishment of consensus-based management and treatment recommendations [30,31,32,33].

In 2014, Walsh et al. introduced a staging system aimed at identifying individuals with a high clinical risk of pulmonary sarcoidosis. This system was developed using a large cohort of patients with pulmonary sarcoidosis, incorporating the composite physiological index and two HRCT variables. The proposed staging system was designed using tests routinely conducted in the majority of patients, making it applicable to individuals with pulmonary sarcoidosis in less selectively chosen populations. A limitation of this staging system is that its validity needs to be improved in populations with less severe disease or at other less specialized centers. The authors integrated prognostic physiologic variables and HRCT to set up a clinical staging algorithm designed to predict mortality in a test cohort. Mortality was the primary outcome. Survival period was calculated starting from the date of baseline pulmonary function tests to the last known point of contact or to date of death. The vital status of all patients was known at the study’s conclusion. By integrating the composite physiological index (CPI) with HRCT measures of pulmonary vasculature and interstitial disease, the researchers proposed a straightforward staging system that identifies patients at high clinical risk. This model should represent a robust prognostic tool for stratifying sarcoidosis patients [34].

In 2020, Baughman et al. made a significant effort to classify the phenotypes of pulmonary sarcoidosis. The Expert Delphi consensus recommended no therapy for asymptomatic patients with normal pulmonary function or adenopathy alone, while symptomatic patients with impaired pulmonary function or infiltrates should be treated. Disagreement exists for asymptomatic patients with abnormal chest images or impaired pulmonary function and for symptomatic patients with normal chest imaging and pulmonary function. Proposed phenotypes include asymptomatic (no therapy), acute (duration <1 year, corticosteroids), chronic (antimetabolites, second-line therapies), and advanced (biologics). Immediate therapy is warranted in specific cases such as dyspnea/hypoxemia at rest, severe impairment in pulmonary function tests, and severe cardiac, neurologic, ocular, or renal involvement. These classifications offer tailored insights into the precise therapeutic management of pulmonary sarcoidosis [35].

### 2.3. Risk Factors

Assessment of risk factors is a pivotal aspect of the diagnostic approach to patients with sarcoidosis. A 2016 study investigated smoking and obesity as potential risk factors in sarcoidosis. The authors concluded that although these factors could influence the risk of sarcoidosis, the data remained inconclusive [36]. While smoking is strongly linked to the development of pulmonary diseases such as chronic obstructive pulmonary disease and lung cancer, intriguingly, previous studies have indicated a lower risk of sarcoidosis associated with smoking. It is worth noting that these studies were conducted using reference cohorts, which are potentially not representative of the true spectrum of the disease [37,38]. Obesity is associated with an increased risk of some autoimmune disorders such as psoriasis and rheumatoid arthritis [39,40]. A recent study utilizing the Black Women’s Health Study cohort revealed an increased risk of sarcoidosis in African-American women with a body mass index (BMI) exceeding 30 kg/m^2^ [8]. However, information on the link between obesity and sarcoidosis risk in other populations remains limited [41]. The authors utilized a population-based cohort to investigate the relationships between smoking, obesity, and sarcoidosis. They observed a negative association between current smoking and the risk of sarcoidosis, which is consistent with the findings from previous referral-based studies in Western countries. The reason for this risk reduction in current smokers remains unclear, suggesting a potential non-causal association influenced by confounding factors. However, considering the known effects of smoking on suppression of T-lymphocyte function and macrophage phagocytic activity, an alternative explanation is that smoking may disrupt the macrophage–lymphocyte activation process, leading to granuloma formation [38,42]. This negative association is in line with the findings in hypersensitivity pneumonitis, which is another granulomatous lung inflammatory disease [43,44]. It is noteworthy that this negative association between smoking and sarcoidosis was not observed in Asian cohorts, suggesting potential regional or population-specific etiopathogenesis [45,46]. Conversely, a positive association between obesity and sarcoidosis risk was identified in the study, particularly among individuals classified as obese, but not among those who were overweight. The precise mechanism linking obesity to sarcoidosis remains unclear, but excessive leptin secretion from adipocytes in obese patients is proposed as a possible explanation. Leptin, a pro-inflammatory adipokine, shows potent immunomodulatory effects; these potentially contribute to autoimmunity and increase the risk of sarcoidosis [47,48].

## 3. Biomarkers

### 3.1. Serum Biomarkers

Due to the intrinsic complexities associated with the diagnosis and clinical management of sarcoidosis, much attention has been paid in recent years to the identification and standardization of biomarkers. The imperative for specific biomarkers characterized by optimal sensitivity and specificity is paramount for prognosticating clinical outcomes and making informed clinical decisions [49]. Numerous serological biomarkers have been documented in the literature for identifying and prognostically assessing sarcoidosis. This section aims to outline the main ones studied, present new evidence, and unveil emerging discoveries in this field. The most relevant biomarkers are summarized in Figure 1.

Serum angiotensin-converting enzyme (sACE) is a matrix metallopeptidase primarily expressed in lung tissue, which is responsible for the formation of angiotensin II (Ang II), a biologically active peptide hormone [50]. Widely used and studied in sarcoidosis, [51] sACE was initially described by Lieberman in 1975 [52]. While employed as a biomarker and for comparative purposes, its diagnostic utility is controversial due to its low sensitivity and specificity. sACE can be accessed via enzymatic activity or protein concentration measurement. Notably, most pathology labs measure ACE activity, which is affected by ACE inhibitor (ACEI) drugs used in hypertension, whereas ACE protein levels remain unaffected [53]. ACEIs erode the negative predictive value of ACE activity, which makes it necessary to examine the use of drugs in patients with sarcoidosis [53,54]. ACE insertion/deletion polymorphism also influences sACE activity, yielding variable specificity in sarcoidosis patients [55]. In light of this information, the specificity among patients with sarcoidosis exhibits a notably wide range [56]. In a Japanese cohort study, significantly higher ACE values were noted in patients compared to normal controls, but no distinctions were found between clinical and pathological diagnosis groups or cardiac and non-cardiac involvement groups. The study reported ACE sensitivity at 21.5% and a negative predictive value (NPV) at 82.1% [57]. In contrast, a 2022 Chinese cohort study emphasized sACE’s superiority over erythrocyte sedimentation rate (ESR) in reflecting disease activity, reporting a sensitivity of 61.4% and a high specificity of 92.9%. Notably, the authors observed no correlation between sACE and ESR in the same study, underscoring sACE’s efficacy in assessing disease activity [58]. Furthermore, a recent investigation revealed that employing an optimized fluorescent kinetic ACE activity method [59] and genotype-dependent reference intervals significantly enhanced the test’s sensitivity while maintaining a high level of specificity [60]. In 2021, a novel ACE fingerprinting approach was introduced, utilizing a panel of mAbs and plasma samples to explore the possibility that elevated ACE levels in patients are due to mutations in the ACE gene rather than being related to sarcoidosis [61].

Another frequently used biomarker in the clinical management of sarcoidosis is serum amyloid-A (SAA), which is found to be elevated in various inflammatory diseases [62]. According to the literature, SAA plays a pathogenetic role in granulomatous inflammation [63]. Expressed by CD68-positive macrophages and giant cells within granulomas, its correlation with CD3-positive lymphocytes links its expression to the local Th-1 response [64]. A thousand-fold increase of SAA plasmatic concentrations should be observed in inflammatory disorder compared to baseline. Interleukin (IL)-1, IL-6 and tumor necrosis factor (TNF)-α, can induce SAA release within 2–3 h. SAA, interacting with toll-like receptor 2 (TLR-2), modulated the immune response mediated by T-helper-1 lymphocyte (Th-1) immune response [65]. SAA can also induce the expression of NF-kB, stimulating TLR-2 expression. Additionally, SAA appears to mediate lipid metabolism in sarcoidosis. A recently published study found an association between reduced serum concentrations of HDL cholesterol in untreated patients with active sarcoidosis and high serum levels of SAA [66]. However, elevated SAA levels are also observed in various inflammatory diseases such as RA and Crohn’s disease [67], as well as several other pulmonary conditions [63,68,69], casting uncertainty on its role as a diagnostic or prognostic biomarker.

Krebs von den Lungen-6 (KL-6) is a mucin-1 glycoprotein encoded by the MUC1 gene [70]. It is normally expressed on type II pneumocytes in lung tissue. Serum KL-6 is thought to be produced by regenerating type II epithelial cells, with levels correlating to disease activity [71]. KL-6 also functions as a cell-surface barrier and should play a protective role associated with damage to the alveolar epithelial barrier molecule [70]. Notably, KL-6 levels have been reported to exhibit an inverse correlation with DLCO [49]. However, various studies suggest that serum KL-6 levels may reflect lymphocytic alveolitis and increased parenchymal infiltration [71]. In the study by Bergantini et al., KL-6 values, below a cut-off of 803.5 IU/mL, demonstrated 72.2% sensitivity and 86.4% specificity in distinguishing sarcoidosis patients from other groups [49].

Chitotriosidase (CTO) is an enzyme classified within the chitinase family, which is tasked with breaking down chitin: a polymer present in the cell walls of fungi and the exoskeletons of insects and crustaceans. Upon activation of Toll-like receptors by interferon-γ (IFN-γ), tumor necrosis factor (TNF), and granulocyte/macrophage colony-stimulating factor (GM-CSF), pulmonary neutrophils and macrophages release this enzyme [72]. A study reported a sensitivity of 83–89% for chitotriosidase [73]. In an Italian study involving 694 sarcoidosis patients, chitotriosidase activity and ACE concentrations show a direct correlation [74]. The same study indicated a significant increase in chitotriosidase activity in cases of severe pulmonary involvement, identified by evidence of lung fibrosis with reticular abnormalities and traction bronchiectasis on HR-CT scan [74]. Similar findings were observed in another study, where high CTO activity demonstrated discriminatory power with a specificity of 100% [75]. Moreover, in a different Italian study, CTO, with a value exceeding 90.86 nmol/mL/h, appeared capable of distinguishing between active sarcoidosis and patients with fever of unknown origin (FUO) without sarcoidosis, with high sensitivity and specificity [96.8% (84.2–99.9) and 85.5% (75.0–92.8), respectively] [76]. However, CTO gene polymorphisms (in particular, a common mutation of the CHIT1 gene) significantly modify CTO activity [60], potentially leading to false-negatives [77].

Carbohydrate antigen (CA) 125 is a high-molecular-weight transmembrane glycoprotein that dissociates from the cell membrane and is released into the serum as circulating CA125. Notably, CA125 expression is heightened in inflamed tissue [78]. Elevated serum CA125 levels have been documented in both malignant and non-malignant diseases [79]. A recent study [80] reported a correlation between CA125 levels and the clinical presentation of sarcoidosis, lung function (FVC% predicted), and Scadding’s radiological classification. The study identified an optimal CA125 cut-off point of 32.33 U/mL, yielding a sensitivity of 96.3% and a specificity of 90.2%. Furthermore, inverse correlations were observed between serum CA125 levels and baseline FVC% predicted [80].

Galectins (Gal) are proteins that bind to β-galactosides; specifically, N-acetylactosamine present in N-linked and O-linked glycoproteins. Several studies demonstrated their role in interstitial lung diseases (ILDs), in which they play a vital role in resolving inflammation by inducing apoptosis in activated leukocytes [81,82]. In a cohort study conducted in Turkey, serum levels of galectin-3 were found to be significantly higher in sarcoidosis patients compared to healthy controls. Furthermore, these levels were particularly elevated in patients at stage 2 or higher of the disease [83].

Neuron-specific enolase (NSE) is a neuron-specific glycolytic isozyme of enolase primarily expressed in neurons and neuroendocrine cells. However, it is also found in non-neural cells such as platelets, lymphocytes, and macrophages [84]. Previous studies indicate that serum NSE levels increase not only in conditions with pulmonary correlations, such as pulmonary tuberculosis [85] or alveolar proteinosis [86], but also in systemic inflammatory disease such as Crohn’s disease [87] and kindly disease [88]. The results of a retrospective study on serum NSE level in sarcoidosis patients suggest that this marker has poor sensitivity on its own but shows improvement in combination with serum levels of ACE and sIL-2R (93.8%) [89]. Moreover, in the past two years, several authors have described new biochemical markers or enhanced knowledge about their pathways and clinical significance. Some studies have delved into the role of tissue growth factors as markers of disease and their prognostic value. A study published in 2022 revealed significantly elevated plasma levels of IL-1RA (Interleukin-1 Receptor Antagonist), eNAMPT (Extracellular Nicotinamide Phosphoribosyltransferase), ANG-2 (angiotensin-2), IL-6, and HBEGF (Heparin-binding EGF-like growth factor) in patients with sarcoidosis, though not IL-8 levels [90]. In the same study, the authors demonstrated that the tissue expression of biomarkers in immunohistochemistry (IHC) assays was largely consistent with the results observed in plasma biomarker assessments. Elevated expression of NAMPT, HBEGF, and ANG-2 was noted in lung and lymph node tissues from individuals with sarcoidosis [90]. In the literature, a correlation between HBEGF and ground glass score in CT-ILDs, is described, but there is no such correlation with fibrosis score [91]. In a separate study involving 54 patients, an association was found between lung function and serum levels of PDGF-AB and VEGF in individuals with sarcoidosis. However, VEGF concentrations did not differ significantly among patients with sarcoidosis, idiopathic pulmonary fibrosis (IPF), or controls [92]. An investigation into B cell-activating factors (BAFFs) in plasma levels conducted on 55 sarcoidosis patients revealed the following (90). Compared with controls, serum BAFF concentration was significantly higher in both active chronic sarcoidosis and acute sarcoidosis groups. Nonetheless, the mean BAFF level in the two patient groups did not exhibit a significant difference.

Isshiki et al. recently highlighted the potential diagnostic role of matrix metalloproteinase 7 (MMP-7), CC-chemokine ligand 18 (CCL-18), and periostin [93]. MMP-7, a zinc-dependent endoprotease, is thought to play a role in the injury-repair process by stimulating cell migration and coordinating the inflammatory response [94]. The study revealed elevated plasma MMP-7 levels in sarcoidosis patients with parenchymal infiltration, correlating with poor lung function [93]. CCL-18, a cytokine produced by alveolar macrophages, stimulates lung fibroblasts to produce collagen [95], and a study suggest that it should be correlated with sarcoidosis activity [93]. Periostin, an extracellular matrix protein, was found to be increased in sarcoidosis patients, and may correlate with ACE and IL-2 serum levels [93].

During the research process, some biomarkers gain importance, while others, which initially suggested a role, lose their clinical relevance. Visfatin, a pro-inflammatory adipokine, showed no statistical difference between sarcoidosis and control groups in a recent 2020 study and exhibited no correlation with other biomarkers [96]. However, in the same year, another study reported high levels of adiponectin in sarcoidosis patients compared to controls [97]. A 2019 study hinted at the potential role of serum albumin levels as a predictive marker for the duration of the disease. Nevertheless, being a retrospective study, it lacked the capacity to establish correlations between albumin levels and other acute-phase proteins, liver, and kidney functions [98].

New approaches have also been established to discover potential biomarkers. Futami et al., by using a combination of non-targeted and targeted proteomics approaches, identified CD14 and lipopolysaccharide-binding protein (LBP) as potential biomarkers of sarcoidosis [99]. Both CD14 and LBP are crucial for recognizing lipopolysaccharide (LPS) by Toll-like receptor (TLR) 4 [100]. Interestingly, in the same study, these new biomarkers appeared to be uncorrelated with other lung diseases such as COPD or ILD, suggesting good specificity for sarcoidosis [99].

Major recently described biomarkers for sarcoidosis are summarized in Table 1.

### 3.2. Immune System and Sarcoidosis

Hypotheses regarding the onset of the disease suggest that the pathogenesis involves an interaction between the immune system and same environmental antigens (Ag). This unidentified antigen has the potential to engage Toll-like receptor (TLR) 2, TLR4, and nucleotide-binding oligomerization domain (NOD)-like receptor (NLR) 2, thereby triggering the activation of the widespread transcription factor Nuclear Factor-Kappa B (NFκB) [103].

Previous studies have identified increased expression levels of TLR2, TLR4, and NOD2 in antigen presenting cells (APCs), including macrophages, B lymphocytes, and dendritic cells from sarcoidosis patients. This increased expression results in a four-fold higher secretion rate of tumor necrosis factor-α (TNF-α) and a 13-fold higher secretion rate of interleukin-1β (IL-1β) upon stimulation of these receptors, leading to a significant escalation in levels of the systemic pro-inflammatory mediators [104].

Considerable evidence also supports the interaction between human leukocyte antigen (HLA) class I and II molecules and the onset of sarcoidosis. A European study identified various associations between acute manifestations of sarcoidosis and HLA-A1 and HLA-B8 antigens, while HLA-B22 was linked to disseminated systemic disease [105]. Those presumed sarcoid agents, through the activation of naive T cells, may lead to the production of Th1 cytokines by CD4+ and CD8+ cells, facilitating granuloma formation by recruiting monocytes and activating macrophages. The bronchoalveolar lavage (BAL) fluid of patients with sarcoidosis typically shows an imbalance toward CD4+ T cells, with a demonstrated ratio of 4:1 between CD4+ and CD8+ T cells [103].

In a 2021 study, D’Alessandro et al. analyzed the frequency and distribution of B and T lymphocyte subsets in both peripheral and alveolar compartments, with a particular emphasis on follicular T helper cells. As suggested by the literature, a higher rate of CD4 and a lower rate of CD8 cells were observed in BAL compared to peripheral blood in sarcoidosis patients. Elevated serum concentrations of cytokines, including IL-2, IL-4, IL-6, IL-10, TNF, and IFN-γ, were also observed in patients compared to controls [106]. Recent discoveries have highlighted that the activation of CD4+ naive T cells, associated with the increased expression of regulatory pathways, indicates the initiation of a persistent inflammatory response with potential autoimmune or infectious origins. Specifically, an upregulation of JAK/STAT, PI3K/AKT, and ERK/MAPK signaling pathways has been observed, suggesting a general activation of T cells in response to a potential antigen, such as that of naive cells. Additionally, signs of dysregulation in multiple regulatory mechanisms have been identified, including a loss of apoptosis mechanisms in CD4+ naive T cells [107]. Furthermore, altered T helper (Th) subsets were identified in sarcoidosis patients. Percentages of Th2 and Th17 cells were higher in serum than in BAL, while percentages of Tregs were lower. CD5-expressing B cells were fewer in peripheral blood than in BAL, a trend also demonstrated in controls and sarcoidosis patients [106] follicular T helper like cells were identified in the lungs of patients with pulmonary sarcoidosis, producing high levels of IFN-g and IL-21 and providing potent B-cell help in vitro. High levels of circulating follicular Th-like cells in peripheral blood were suggested as a biomarker for monitoring sarcoidosis disease activity [108].

The role of chemokine and monokine has also been investigated in recent years. One recent study by Ragusa in 2018 has reported high levels of Th1 dependent chemokines, MIG, CXCL9, and CXCR3 positive alveolar macrophages in BAL and biopsy samples of sarcoidosis patients. In this study, the epithelioid and giant cells of the sarcoid lungs stained positive for MIG, I-TAC, and IP-10 [109]. Monokine induced by IFN-γ (MIG), also known as CXCL9, is a T-cell chemoattractant. Several studies reported an increase in MIG and other Th1-associated serum levels, as well as their elevation in various organs of specific autoimmune diseases. Increased levels of MIG and CXCR3 were demonstrated in biopsy and BAL fluid samples of patients with sarcoidosis, which is correlated with the number of CD4+ and total lymphocytes. Positive MIG expression was observed in the epithelioid and giant cells in the sarcoid lungs [110].

In recent years, the potential role of the IL-2 receptor as a biomarker for sarcoidosis has become increasingly evident. Interleukin 2 (IL-2) is a crucial cytokine in the immune system, regulating protective immunity and maintaining immune tolerance through CD4+ regulatory T lymphocytes (Treg). Elevated blood soluble IL-2 receptor (sIL-2R) levels have been observed in various human diseases, including autoimmune and inflammatory conditions, solid cancers, hematological malignancies, and infections [111]. A high serum sIL-2R level demonstrates high sensitivity in diagnosing sarcoidosis. Moreover, elevated sIL-2R levels at the time of diagnosis have been predictive for the development of chronic sarcoidosis. However, sIL-2R is not specific to sarcoidosis, as it can also be elevated in patients with idiopathic pulmonary fibrosis (IPF) and chronic hypersensitivity pneumonitis (cHP). Additionally, increased levels have been found in tuberculosis and lymphoma. Specifically, sIL-2R levels above 4700 U/L were predictive for developing chronic sarcoidosis, and these elevated levels were also predictive for the need for systemic therapy [112]. Sunaga et al., in 2022, reported a sensitivity of 84,2% and a specificity of 53,6% (cut-off value 482 U/mL) [89]. A 2019 study demonstrated that serum sIL-2R exhibits high sensitivity and specificity compared to angiotensin-converting enzyme (ACE) in diagnosing sarcoidosis. Therefore, in a group of patients under evaluation for sarcoidosis, serum sIL-2R can effectively discriminate between those with and those without sarcoidosis. [113]. From a prognostic perspective, a retrospective analysis in 2020 revealed that sIL-2R remains the most powerful parameter for predicting spontaneous remission of the disease. In particular, when sIL-2R values exceed 1129.5 IU/mL in the stage I subgroup or 1026.5 IU/mL in the stage II subgroup, the risk of non-spontaneous remission increases [114]. Furthermore, elevated serum sIL-2R levels were significantly linked to poorer clinical outcomes in cardiac sarcoidosis. Notably, the levels of sIL-2R were correlated with disease activities in lymph nodes rather than in the myocardium, as assessed by 18F-FDG PET/CT [115].

In a recent bioinformatics analysis based on gene transcriptome analysis of 303 sarcoidosis samples and 400 normal controls, the Weighted Gene Co-expression Network Analysis (WGCNA) identified two modules highly correlated with sarcoidosis. The first one, encompassing 493 genes, highlighted the significance of the defense response to the virus and innate immune response in the pathogenesis of sarcoidosis. The second one, consisting of 684 genes, revealed a strong association of non-coding RNA (ncRNA) processes, including processing, metabolic activities, degradation, and immune cells, with sarcoidosis through enrichment analysis [116]. Recently, the same genetic variance was associated with phenotypes described by Shupps et al. in 2017 [29]. The abdominal phenotype was associated with rs5007259 at BTNL2 (*p* = 0.003; OR = 0.47) and rs454078 IL1RN (*p* = 0.006; OR = 2.05), and with rs2233409 near NFKBIA (*p* = 0.004; OR = 0.18) and rs2891 near ITGAE/NCBP3 (*p* = 0.005; OR = 0.19). The ocular–cardio–cutaneous–CNS phenotype showed associations with rs12793173 near LOC102723568 (*p* = 0.006; OR = 0.68) and rs3775291 in the TLR3 gene, resulting in a missense variant (*p* = 0.001; OR = 1.59). The extrapulmonary phenotype was associated with rs2015086 near CCL18 (*p* = 0.002; OR = 4.51), rs3800018 near RAB23 (*p* = 0.002; OR = 6.51) and rs7756421 near ZNF451 (*p* = 0.002; OR = 6.53) [117]. The neutrophil/lymphocyte ratio (NLR) has recently gained recognition as a cost-effective and useful inflammatory marker with diagnostic and prognostic value in various respiratory and cardiac diseases. In a recent study, it was shown that higher NLR values at the time of diagnosis were associated with a more advanced disease stage and an active clinical status in individuals with biopsy-proven sarcoidosis. A defined NLR cut-off value of ≥2.39 (with 72.0% sensitivity and 52.0% specificity) was established to effectively discriminate active from stable disease [118].

Macrophages play a pivotal role in the pathogenesis of sarcoidosis, with granulomas comprising cells derived from macrophages. Investigating this cell population is crucial for the clinical management of this intricate disease. A high density of CD163 (+) cells surrounding epithelioid granulomas is linked to systemic involvement in sarcoidosis patients. CD163, a valuable marker for M2 macrophages, is upregulated by IL-6, IL-10, and glucocorticoids, and downregulated by proinflammatory cytokines like tumor necrosis factor-α and interferon-β [119]. The existing literature suggests that hypoxia, a probable microenvironmental factor present in granulomas and inflammatory tissues, triggers the activation of hypoxia-inducible factor (HIF-1). This activation leads to a mixed proinflammatory-profibrotic phenotype in MD-macrophages, which is particularly noticeable in cases of active sarcoidosis. In vitro findings are supported by immunohistochemistry data, demonstrating the expression of HIF-1a and its target PAI-1 in epithelioid cells forming pulmonary sarcoidosis granulomas. This activation may contribute to the development and persistence of granulomas in active sarcoidosis, promoting fibrosis around granulomas through a combined inflammatory and fibrosing macrophage response. Consequently, this diminishes their antigen presentation capacities, resulting in an impaired T cell response. Therefore, the HIF pathway and PAI-1 may play a role in the pathogenesis of highly active sarcoidosis, and reveal targets for future medications [120].

## 4. Conclusions

Sarcoidosis remains a diagnostic challenge. Phenotyping patients is imperative for clinicians in order to try to standardize several crucial points not addressed by current guidelines: to be aware of recurrence, to enhance confidence in dosing and duration of treatment, and to measure response to treatment, as examples. Recognizing the diverse presentations including multi-organ involvement, variable severity, atypical manifestations, uncommon symptoms, extrapulmonary involvement, silent disease, and variable course, is crucial for timely diagnosis and appropriate management of sarcoidosis. The process of phenotyping is dynamic and adapts with the emerging discovery of new disease characteristics, novel biomarkers, and genetic characterization. In conclusion, this paper underscores the potential of serum biomarkers as valuable tools in the diagnosis, prognosis, and monitoring of sarcoidosis.

## 5. Future Directions

Through elucidating the intricate interplay between various biomarkers and disease pathogenesis, the final goal is to move closer to personalized medicine and improved patient outcomes in the management of sarcoidosis, by correct phenotyping of the disease and the recognition of valid serum biomarkers. However, further research is warranted to validate and standardize these biomarkers for routine clinical use, ultimately enhancing our ability to tailor therapeutic strategies and optimize patient care in this complex and heterogeneous disease.

## Figures and Tables

**Figure 1 diagnostics-14-00709-f001:**
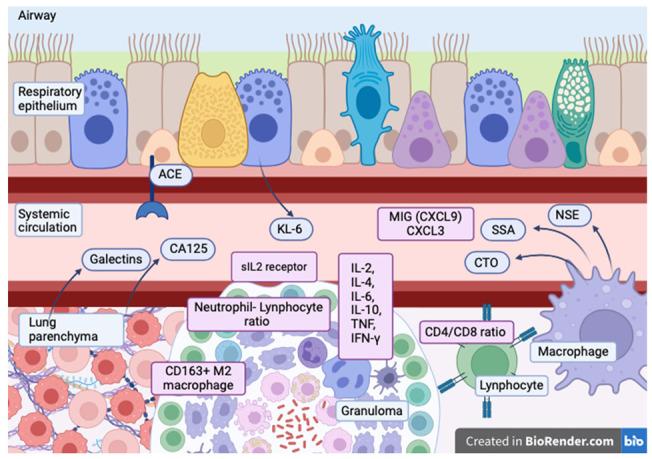
Main serum biomarkers of pulmonary sarcoidosis: angiotensin-converting enzyme (ACE), Cancer Antigen 125 (CA125), galectins, Krebs von den Lungen-6 (KL-6), interleukin-2 (IL-2), interleukin-4 (IL-4), interleukin-6 (IL-6), interleukin-10 (IL-10), serum interleukin-2 receptor (sIL-2 receptor), tumor necrosis factor-alpha (TNF-α), interferon-gamma (IFN-γ), monokine induced by interferon-gamma (C-X-C motif chemokine 9) (MIG (CXCL9), C-X-C motif chemokine 3 (CXCL3), neuron-specific enolase (NSE), serum amyloid-A (SSA), chitotriosidase (CTO), CD163 M2 macrophage.

**Table 1 diagnostics-14-00709-t001:** Sensitivity and specificity of the major serum biomarker for sarcoidosis, according to the most recent literature.

Biomarker	Short Definition	Origin	Cut off Value	Sensitivity (%)	Specificity (%)	Reference
Serum angiotensin-converting enzyme (sACE)	A matrix metallopeptidase responsible for angiotensin II (Ang II) formation.	Lung tissue	21.4 U/L	41.1–61.5	92.5–100	[57,58,89]
Serum Amyloid-A (SAA)	Pathogenetic role in granulomatous inflammation.	CD68-positive macrophages and giant cells within granulomas	Positive	84	44	[101]
Krebs von den Lungen-6 (KL-6)	Mucin-1 glycoprotein	Type II pneumocytes	Below 803.5 IU/mL	72.2	86.4	[49]
Chitotriosidase (CTO)	Enzyme with role as chitinase	Pulmonary neutrophils and macrophages	>90.86 nmol/mL/h	96.8	85.5	[76]
Carbohydrate antigen (CA) 125	High-molecular-weight transmembrane glycoprotein	Inflamed tissue	32.33 U/mL	96.3	90.2	[80]
Galectins (Gal)	Proteins that bind to β-galactosides inducing apoptosis in activated leukocytes	Monocytes, macrophages, endothelial and epithelial cells	17.96 ng/mL	88.9	85.7	[102]
Neuron-specific enolase (NSE)	Neuron-specific glycolytic isozyme of enolase	Neurons and neuroendocrine cells, platelets, lymphocytes, and macrophages	12.0 ng/mL	50.9	74.2	[89]
Plasma Matrix Metalloproteinass -7 (MMP-7)	Zinc dependent endoprotease	Lung epithelium, macrophages	3.7 ng/mL	80	70	[93]
CC-Chemokine ligand 18 (CCL-18)	Stimulation of lung fibroblast to produce collagen	Alveolar macrophage	52.3 ng/mL	63.3	64.7	[93]
Periostin	Extracellular matrix protein	Several tissue	72.5 ng/mL	73.3	70	[93]
CD14	Recognizing lipopolysaccharide by Toll-like receptor	Monocyte antigen	NA	NA	NA	[99]
Lipopolysaccharide-binding protein (LBP)	Recognizing lipopolysaccharide by Toll-like receptor	Monoclear cells in lung and lymph note	NA	NA	NA	[99]
IL2R	Interleukin 2 receptor	Activated LyT cells and macrophages	482 U/L	84.2	53.2	[89]

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
