# Peer review of "Phenotypes and Serum Biomarkers in Sarcoidosis"

_diagnostics, 2024, doi:10.3390/diagnostics14070709_

Round 1

Reviewer 1 Report

Comments and Suggestions for Authors

In this paper, the authors summarized different biomarkers in sarcoidosis and provided a good overview with many references. It will be interesting for both clinicians and researchers.

I have the following comments:

Abstract

Almost half of the abstract is used for future perspectives (lines 21-26). Could it be reduced or maybe removed?

Introduction

Line 47: 10% of patients need systemic treatment. I find 10% extremely low; please verify this with more references. This may be the case in pulmonary sarcoidosis, but not when you include all the patients with neurosarcoidosis, cardiac sarcoidosis, hypercalcemia, etc.

Systemic disease

Maybe the authors can use the following papers:

Lhote R, Annesi-Maesano I, Nunes H, et al. Clinical phenotypes of extrapulmonary sarcoidosis: an analysis of a French, multi-ethnic, multicentre cohort. Eur Respir J 2021; 57: 2001160 [https://doi.org/10.1183/13993003.01160-2020].

Rubio-Rivas M, Corbella X. Clinical phenotypes and prediction of chronicity in sarcoidosis using cluster analysis in a prospective cohort of 694 patients. Eur J Intern Med. 2020 Jul;77:59-65. doi: 10.1016/j.ejim.2020.04.024. Epub 2020 Apr 21. PMID: 32331839.

Schupp JC, Freitag-Wolf S, Bargagli E, et al. Phenotypes of organ involvement in sarcoidosis. Eur Respir J 2018; 51: 1700991 [https://doi.org/10.1183/13993003.00991-2017].

Pulmonary disease and Risk factors

Excellent

Serum biomarkers

This is a complicated area. I suggest adding a table to the overview (e.g., source, sensitivity, specificity) and highlighting the most relevant biomarkers in a clinical context.

Immune system and sarcoidosis

In this section, the authors give a quick overview of several extremely complicated areas. This is understandable, given the length of the article.

As IL2R is the most clinically relevant biomarker, I propose to add the exact figures for specificity and sensitivity. I also suggest adding IL2R to the table of serum biomarkers.

In the genetic section, some of the most significant genes should be mentioned.

Finally, I missed a section about PET-CT.

Author Response

Thanking reviewer 1 for the valuable suggestions and the right improvement to the text, we report below the corrections made in the text:

Introduction:

Lines 46-47: We have modified the sentence regarding the percentage of patients requiring systemic treatment, considering the publications suggested by the reviewer. The revised sentence reads, "Patients should require systemic treatment in different proportions ranging from 10 to 73%, in different cohorts."

Systemic Disease:

Lines 92-97: We have added a paragraph considering the phenotypic classification suggested in the third article cited by the reviewer. The paragraph reads: "In 2018, a multicentric study published in the European Respiratory Journal proposed a classification of systemic sarcoidosis identifying 5 clusters of patients based on organ engagement: I) Abdominal, II) ocular-cardiac-cutaneous-CNS, III) musculoskeletal-cutaneous IV) pulmonary-Lymph nodal V) Extrapulmonary Sarcoidosis. According to the authors, this classification should provide a better definition of the subcohort of patients with sarcoidosis."

Serum biomarkers:

We agree with the reviewer on the need to add a table to make the data from the review easier and clearer to understand. We have integrated the comments from both reviewers into this table. It is located at line 343.

Immune system and sarcoidosis:

Lines 430-439: We have cited some of the genetic mutations recently associated with sarcoidosis phenotypes from a recent European multicentric study, following reviewer’s advice.

Regarding the absence of a paragraph about the role of PET-CT, we agree with the reviewer on the importance of this technique in staging and studying the Sarcoidosis's progression. However, we believe that the topic requires a broader discussion than what we could provide in this paper. Adding a single paragraph might appear disjointed from the theme of the work and insufficient in covering the topic comprehensively.

Once again, we thank the reviewer for their courtesy and attention to our work.

Please see the attachment for the new version of the review.

Best regards.

Reviewer 2 Report

Comments and Suggestions for Authors

The manuscript is well organized,innovative, written in standard English. It offers a summary of the last 10 years of a very specific lung disease - sarcoidosis. No specific comments on the layout or style. 

Suggestions:

1. the authors to add a pattern of specificity in lung lobe involvement;

2. to be add a graphic model of the information presented after each subsection - this will make it very easy to understand due to the specificity of the subject matter.

3. The concluding section should be modified and emphasis should be placed on innovativeness of disease presentation. 

Comments on the Quality of English Language

Minor editing of English language required

Author Response

We thank reviewer 2 for their timely correction of our work, and we will address their suggestions accordingly:

  1. Regarding suggestion 1, we believe that the topic is indeed very interesting for studying sarcoidosis. However, it would require an extensive and in-depth discussion that may not align well with the focus of our current work, which is more centered on serum biomarkers. It could be an excellent topic for future paper.
  2. We agree with the reviewer on the need to add a graphical support to make the paragraph on biomarkers more accessible. Therefore, we have added a table at line 343, incorporating the suggestions from both reviewers.
  3. Lines 468-472: In the conclusions, following reviewer’s suggestion, we have included the following sentence: "Recognizing the diverse presentations including multi-organ involvement, variable severity, atypical manifestations, uncommon symptoms, extrapulmonary involvement, silent disease, and variable course, is crucial for timely diagnosis and appropriate management of sarcoidosis."

We thank the second reviewer for their valuable contribution.

Please see the attachment for the new version of the review.

Kindly regards.
